# The Role of *Medicago lupulina* Interaction with *Rhizophagus irregularis* in the Determination of Root Metabolome at Early Stages of AM Symbiosis

**DOI:** 10.3390/plants11182338

**Published:** 2022-09-07

**Authors:** Andrey P. Yurkov, Roman K. Puzanskiy, Alexey A. Kryukov, Anastasiia O. Gorbunova, Tatyana R. Kudriashova, Lidija M. Jacobi, Andrei P. Kozhemyakov, Daria A. Romanyuk, Ekaterina B. Aronova, Galina S. Avdeeva, Vladislav V. Yemelyanov, Alexey L. Shavarda, Maria F. Shishova

**Affiliations:** 1Laboratory of Ecology of Symbiotic and Associative Rhizobacteria, All-Russia Research Institute for Agricultural Microbiology, Pushkin, 196608 St. Petersburg, Russia; 2Graduate School of Biotechnology and Food Science, Peter the Great St. Petersburg Polytechnic University, 194021 St. Petersburg, Russia; 3Faculty of Biology, St. Petersburg State University, 199034 St. Petersburg, Russia; 4Laboratory of Analytical Phytochemistry, Komarov Botanical Institute of the Russian Academy of Sciences, 197376 St. Petersburg, Russia; 5Center for Molecular and Cell Technologies, St. Petersburg State University, 199034 St. Petersburg, Russia

**Keywords:** *Medicago lupulina*, arbuscular mycorrhiza, *Rhizophagus irregularis*, symbiotic efficiency, plant development, physiological stage, root, metabolic profile

## Abstract

The nature of plant–fungi interaction at early stages of arbuscular mycorrhiza (AM) development is still a puzzling problem. To investigate the processes behind this interaction, we used the *Medicago lupulina* MlS-1 line that forms high-efficient AM symbiosis with *Rhizophagus irregularis*. AM fungus actively colonizes the root system of the host plant and contributes to the formation of effective AM as characterized by a high mycorrhizal growth response (MGR) in the host plant. The present study is aimed at distinguishing the alterations in the *M. lupulina* root metabolic profile as an indicative marker of effective symbiosis. We examined the root metabolome at the 14th and 24th day after sowing and inoculation (DAS) with low substrate phosphorus levels. A GS-MS analysis detected 316 metabolites. Results indicated that profiles of *M. lupulina* root metabolites differed from those in leaves previously detected. The roots contained fewer sugars and organic acids. Hence, compounds supporting the growth of mycorrhizal fungus (especially amino acids, specific lipids, and carbohydrates) accumulated, and their presence coincided with intensive development of AM structures. Mycorrhization determined the root metabolite profile to a greater extent than host plant development. The obtained data highlight the importance of active plant–fungi metabolic interaction at early stages of host plant development for the determination of symbiotic efficiency.

## 1. Introduction

The efficient symbiosis between higher plants and arbuscular mycorrhiza (AM) fungi is one of the more intriguing questions in modern biology. AM is a widespread symbiosis formed by more than 80% of land plants and Glomeromycetes fungi [1]. AM plays an important role in terrestrial ecosystems. Such plant–fungal association enhances plant growth and its adaptive capabilities [1,2,3]. This phenomenon has both biological and agricultural importance, as environmental stresses can reduce crop production by up to 70% [4]. Nevertheless, the lack of mycorrhizal growth response (MGR) was shown for *Pisum sativum*, *Plantago lanceolata, P. major, M. truncatula, Veronica chamaedrys, Poa annua,* and *Vitis vinifera* inoculated with *R. irregularis* [5,6,7]. Thus, the importance of determining new parameters that reflect the intensity of interactions between AM partners is self-evident. 

The intensification of AM biology research is based on the employment of new so-called “omic” technologies. AM fungi effect the transcriptome of *M. truncatula, Nicotiana attenuata, Sorghum bicolor* [8,9,10], and the proteome of *M. truncatula* and *Sorghum bicolor* [11,12,13,14]. Special attention was paid to AM manifesting metabolic rearrangements [5,15,16,17]. Its significant species specificity has been revealed for plants [6,18,19,20,21,22]. This was demonstrated for a number of model plants: *Pisum sativum* L. [23,24,25], *Medicago truncatula* Gaertn. [26,27], *M. lupulina* L. [17,28], *Vicia faba* L. [23], *Lotus japonicus* L. [29], *Phaseolus vulgaris* L. [30], *Zea mays* L. [31], *Solanum lycopersicum* L. [32], *Petunia×hybrida* hort. ex Vilm. [33], etc. At the same time, the mycosymbiont, being an obligate symbiotroph, receives photosynthates from the host plant [34] and is known to synthesize a number of specific metabolites [35,36,37]. Nevertheless, AM fungi species specificity has not yet been truly confirmed. In more than half of studies, *Rhizophagus irregularis* strains were used to elucidate the effect of AM fungi on the diversity of primary and secondary metabolites of the host plant [16]. AM fungi induce a wide range of interactions from mutualistic to parasitic [38]. The highest symbiotic efficiency is manifested under low phosphorus (P) level in the substrate. The contribution of inorganic P absorption due to AM fungi through hyphae and arbuscules can reach 90–99% of the total uptake of the host plant [39,40]. Under conditions of P deficiency, the plant–fungal interaction triggers a shift in the profile of primary metabolites: a decrease in leucine, isoleucine, phenylalanine, aspartic acid, tryptophan, and tyrosine in the roots of *Solanum lycopersicum* and an increase in glutamic acid [41], as well as an increase in the total content of proteins and carbohydrates in the leaves of *Anadenanthera colubrina* [42]. Analysis of the mycorrhization effect of *R. irregularis* (previously ascribed erroneously to *Glomus intradices*; [43]) on *M. truncatula*’s metabolism under conditions of P deficiency revealed higher levels of trehalose, asparagines [27], aspartic acid, glutamic acid [6,27], oleic acid, palmitic acid, palmitvaccenic acid, vaccenic acid [27], homoserine, isoleucine, ornithine, phenylalanine, serine, and threonine [6]. On the contrary, P excess coupled with *R. irregularis* inoculation prompted lower levels of asparagine, fatty acids, and their esters, glutamine, phenylalanine, and alanine but a higher level of xylitol in *Triticum durum* plants [44]. Besides AM-induced alterations in the primary metabolism of host plants, a number of studies indicate an increase in the content of a number of secondary metabolites: apocarotenoids, especially blumenols [16,38,45,46,47], mycorradicins [45], cyclohexenone derivatives [27], and abscisic acid [48]; isoflavonoids such as daidzein, malonylononin and ononin [27]; phenylpropanoids, tomatine [15], and other secondary metabolites [16] and species-specific metabolites such as glabridin, withaferin-A, alpha terthienyl [49], and steviol glycosides [50]. 

Revealed metabolic alterations commonly reflect the sum of ongoing metabolic changes specific to a plant’s organs and tissue. These are also prompted by both host plant development and establishment of AM symbiosis. Thus, the detection of time points for the metabolic profile analysis is crucial. Usually, this is calculated as the number of days after inoculation (dai). Most studies related to the plant metabolome were provided once at late stages of AM symbiosis: analysis of plant metabolites at 180 dai in *Glycyrrhiza glabra* [49]; 127 dai in *Leymus chinensis* [51]; 90 dai in *Withania somnifera* and *Tagetes erecta* [49]; 85 dai in *Elymus nutans* and *E. sibiricus* [52]; 84, 77, and 70 dai in *Lotus japonicus* [53]; 70 dai in *Senecio jacobaea* [47]; 62 dai in *Plantago lanceolata*, *P. major*, *Veronica chamaedrys*, *M. truncatula*, and *Poa annua* [6]; 60 dai in *Vitis vinifera* [7]; 50 dai in *Solanum lycopersicum* [15]; ~50 dai in *Lactuca sativa* [54]; 49 dai in *Eclipta prostrate* [55]; 42 dai in *Medicago truncatula* [56]. 

Only limited investigations were focused on plant metabolomes throughout development. As an example, the analysis of *Stevia rebaudiana* leaves at 69, 89, and 123 dai [50]; *P. sativum* leaves and roots from 7 to 110 dai at six stages of the pea plant development [5,25]; *M. lupulina* leaves from 14 to 52 dai at seven stages of black medic growth [17]. Moreover, a literature review indicates that authors do not commonly associate the selected dai with a specific transition of the host plant to a new developmental stage and/or the intensification of AM symbiotic invasion. This makes it difficult to compare the results obtained over different studies. Special attention is merited by the rare data obtained at early stages of AM symbiosis development because this is the period when AM fungus is actively colonizing the host plant’s root system [27,48]. These results can contribute to discovering the mechanisms that appear at a biochemical level over the intense interaction of symbiotic partners. Among such investigations are: *Anchusa officinalis* stems at 9 dai [57], *M. truncatula* leaves at 28 dai [40], *Sorghum caudatum*, and *S. bicolor* roots at 30 dai [38].

This study aims to uncover the metabolic response of *M. lupulina* roots in mycorrhization with *R. irregularis.* For research purposes, a highly responsive *M. lupulina* line, MlS-1 (up to MGR > 350% depending on growth conditions), was selected see [17,28,58]. This line expresses the dwarf phenotype under conditions of low P level in the substrate, a deficiency which is significantly reduced owing to mycorrhization. The AM effect on the metabolic profile of *M. lupulina* roots was detected with GS-MS at early stages of plant growth (14 and 24 dai) under conditions of P deficiency. This period was characterized by active root mycorrhization. The results obtained clarified the intensity of metabolic response triggered by both partners: the host plant’s early development (stages of the first and second leaf) and AM symbiosis. The assumption is that the root metabolic profile alterations caused by mycorrhization are prioritized over those that result from the host plant growing.

## 2. Results

### 2.1. Medicago lupulina MlS-1 Line Plants Development under Low Phosphorus Level

The results of comparative analysis of *Medicago lupulina* plants are presented in Figure 1. The study was performed at the early development stages of *M. lupulina* plants and AM symbiosis with the *Rhizophagus irregularis* fungus under low P level in the soil. At the 14th day after sawing and inoculation (DAS), both plants that control for without AM (“AM−”) and the inoculated variant (“AM+”) had one real leaf. After another week, at 24th DAS, the plants showed differences in the number of leaves: 2.27 ± 0.04 leaves of “AM−” plants and 2.58 ± 0.07 leaves (MGR in the number of leaves = +14%) of “AM+” plants. At 24th DAS, the aboveground parts’ growth intensification was detected in “AM+” plants in comparison to the “AM−” control, calculated by both fresh (Figure 1A) and dry weight (Figure 1B). However, MGR as expressed in fresh and dry root weight was absent.

Further analysis showed that, despite the absence of a clear host plant response, mycorrhizal infection was actively developed in the roots (Appendix A). At 14 DAS, all symbiotic structures were observed: arbuscules, vesicles, and intraradical mycelium, and the formation of appressories on the root surface (Appendix A). One week later, the formation of all symbiotic structures was well traced, arbuscules also occupied more than half of the mycorrhized parts of the roots, and active development of the intraradical mycelium was observed (Appendix A). Thus, there were no significant changes in the mycorrhization parameters over the tested period, excepting a decrease in the abundance of vesicles in the mycorrhized part of the roots (Appendix A). The presented data indicate that intensive mycorrhization has occurred at the early stages of host plant and symbiotic development.

### 2.2. General Characteristics of Metabolic Profiles

We analyzed the root metabolic profiles of AM+ and AM− *M. lupulina* plants. The obtained metabolic profiles (Appendix A) included 316 metabolites, of which 79 compounds were identified precisely; only the class was determined for 53 more compounds, and the rest could not be determined from the databases used. Sugars were the most widely represented in the obtained profiles (54), including pentoses, hexoses, and their derivatives, such as sugar alcohols and sugar acids. Some compounds were annotated as “complex sugars” and included those whose molecules contain sugar residues. Among them, different di- and trisaccharides and their derivatives were probably most abundant. Twenty-eight amino acids (including all proteinogenic ones, except arginine), about a dozen carboxylic acids, intermediates of energy metabolism, 11 fatty acids and their derivatives, as well as amines, nitrogenous bases, sterols, etc., were also detected.

Primarily, a comparison of the metabolic profiles of the roots of mycorrhized and non-mycorrhized black medic (*M. lupulina*) plants at different development stages was given a principal component analysis (PCA) (Figure 2A) and multidimensional scaling (MDS) (Appendix A) in a low-dimensional space. The profiles are grouped according to both the age of the host plant and mycorrhization. In the case of PCA, the differences between AM+ and AM− are related to the first PC, accounting for 31% of the variance. Mycorrhized plants at both tested periods were located close to each other in the PCA plot and do not differ in the first PC1 but do differ in PC2. On the other hand, the differences between 14 and 24-day-old AM− plants are associated with PC1. Thus, the metabolic development of the AM+ and AM− roots differs strongly over the tested period. Hierarchical clustering (Figure 2B) shows that samples are grouped primarily according to mycorrhization and secondly by age. Thus, we can conclude that mycorrhization is more decisive in determining the metabolite profile than is the age.

### 2.3. The Effect of Mycorrhization at 14th DAS

To identify differently accumulated metabolites (DAMs) in response to mycorrhization, we used OPLS-DA. The obtained OPLS-DA model included one orthogonal component: R^2^Y = 0.99 (*p* = 0.002), Q^2^Y = 0.94, (*p* = 0.002). The predictive component associated with the status of mycorrhization accounted for 29% of the variance. In Figure 3, the results are visualized on a graph where the nodes correspond to metabolites and the edges correspond to the reactions in which one compound transforms into another. In two-week-old plants, mycorrhization contributed to the accumulation of certain amino acids, leucine, glycine, histidine, tyrosine, tryptophan, and lysine, but was followed by an increase in ornithine, which plays an important role in nitrogen metabolism. In contrast, asparagine and proline levels decreased during mycorrhization. Similarly, intermediates of the Krebs cycle (citrate, succinate, malate, and fumarate) were reduced. In contrast, the level of aconitate increased. The obtained data indicate a possible repression of the TCA cycle at early stages of mycorrhization. AM fungi development had little effect on the profile of monosaccharides, and only a few of them showed significant multidirectional differences. On the contrary, the group of “complex sugars” had a number of DAMs, demonstrating increased levels during mycorrhization. Similarly, the roots of AM+ plants contained a higher representation of sterols. The effect of AM fungi on the profile of free fatty acids was weak. A decrease in linoleic and linolenic acids was detected as well as an increase in an unidentified acylglycerol.

In order to identify the biochemical pathways most susceptible to mycorrhization, we provided a set enrichment analysis (SEA). We used the loadings of the predictive component from the OPLS-DA model as a ranking statistic and sets of metabolites for pathways obtained from the KEGG database. Some tendencies toward the repression of the citrate cycle (TCA cycle) (*p* = 0.016), as well as stimulation of steroid biosynthesis (*p* = 0.005), were revealed.

### 2.4. The Effect of Mycorrhization at 24 DAS

The OPLS-DA model as built included one orthogonal component: R^2^Y = 1.00 (*p* = 0.003), Q^2^Y = 0.97, (*p* = 0.003). At this stage (24 DAS), 45% of the variance was associated with the effect of mycorrhization. This is 1.5 times higher than that at the previous stage. AM fungi stimulated more intensive accumulation of metabolites (Figure 4). Amino acids, including histidine, threonine, beta-alanine, cysteine, methionine, and serine, were among them. To the contrary, levels of oxoproline and asparagine observed a certain downward trend.

It must be noted that, at 24th DAS, the repressive effect of mycorrhization on the metabolites of the Krebs cycle weakened. The citrate level was still reduced, while the downregulation of succinate and malate became less pronounced. The level of oxalate, metabolically associated with TCA cycle, became higher at this stage, as well as the aconitate level. AM+ plants had a greater accumulation of C3 compounds associated with glycolysis: glycerate and glycerol. At 24 DAS, mycorrhization strongly influenced the profile of monosaccharides. It positively regulated the level for most pentoses and hexoses, including glucose and fructose. The effect on “complex sugars” remained but became multidirectional. The stimulation of sterols also continued. At 24 DAS, in contrast to previous the time point, mycorrhiza had strong a negative effect on the accumulation of most free fatty acids. These observations were confirmed by SEA, which showed (Figure 5) repression of the pathways associated with the exchange of fatty acids and the activation of the carbohydrate metabolism.

The comparison of the AM fungi effect on root metabolite profiles at two time points was realized with a SUS plot (shared and unique structures), where metabolites were scattered in the space of factor loadings from the two OPLS models described above (Figure 6). One can see that most metabolites showed the same signs of loading, and a correlation between them was observed (r_s_ = 0.48, *p* < 10^−16^). This indicates a similarity of influence. The common features of the effect at 14 and 24 DAS include an increase in the level of “complex sugars”, amino acids, and nitrogen-containing compounds. However, the correlation coefficient is low, and a large number of metabolites reveal noticeable differences in the value of loadings. However, the specificity of mycorrhization effects at 24 DAS was expressed in a higher accumulation of monosaccharides, some carboxylates, and a decrease in the level of several amino acids.

### 2.5. The Effect of Mycorrhization on Metabolite Allocation between Leaves and Roots

The root of the plant is one of the main sink organs for metabolites. An intensive redistribution of carbon in and between plant organs is supposed and may be affected by AM development. To determine the role of mycorrhization at early stages, we compared, at first, the metabolic profiles of roots and leaves and elucidated how mycorrhization interferes in these differences. PCA analysis (Figure 7A) showed that the metabolite profiles of roots and leaves are very diverse. The differences, as defined, significantly exceed the influence of both the stage of development and the mycorrhization. The OPLS-DA performed to compare roots and leaves showed that the differences between organs are quite similar in AM+ and AM− plants (for loadings r_s_ = 0.77, *p* = 10^−15^). These differences are primarily associated with a large accumulation of amino acids and nitrogen-containing compounds in the roots (Figure 7B). Nonetheless, some metabolites exhibited differences in the patterns of dissimilarity between accumulation in roots and leaves as contingent on AM. This concerns amino acids and some carboxylates. This accumulation in roots in comparison with leaves might be controlled by mycorrhiza. On the other hand, AM fungi development depresses the pool of fatty acids and glycerol in roots as compared to leaves.

To determine the specificity of mycorrhiza effects on the leaves and roots of black medic, we compared the factor loadings of predictive components of OPLS-DA models for the classification of the mycorrhized and control plants (Figure 7C). It was shown that, at 14 DAS, there is a significant positive correlation between factor loading from models for roots and leaves (r_s_ = 0.49, *p* = 10^−5^). However, at 24 DAS, this phenomenon disappeared. Thus, the common scheme of mycorrhiza-induced alterations in roots and leaves is observed only at the initial stage of host plant and AM development.

Thus, the mycorrhization effect on the metabolomics profiles of leaves and roots can be determined by the biochemical and physiological features of organs, as well as the particular nature the of mycorrhization effect’s development. To reveal these particularities, we compared the loadings of the corresponding OPLS-DA models. However, no correlation was obtained between them. Nevertheless, nitrogen-containing compounds and amino acids were characterized by a higher content in roots compared to the leaves (Appendix A) and as such were increased during mycorrhization.

## 3. Discussion

Phosphorus (P) deprivation is one of the most widespread environmental stressors that plants are faced with over the course of their growth. To elucidate the mechanisms as to how plants adapt to P starvation is of a great importance. It was clearly shown that such mechanisms could be detected at transcriptomic and proteomics levels and could easily be phenotypically distinguished by an increase in root architecture and root/shoot ratio elevation [59,60]. In our study, we used a special *M. lupulina* line, MlS-1, which demonstrated a dwarf phenotype under a condition of low P level in the substrate. It was of an interest as to how this black medic would behave at early stages of development, namely at first and second leaf. This period is characterized by the appearance of new leaves. This stage, of course, facilitates photosynthesis, but it has not yet been determined how it affects root development under conditions of limited nutrients. Our results indicated that P deprivation, even at early stages of development, facilitated the accumulation of root fresh weight and almost double the increase of root/shoot ratio (Figure 1). These data are in agreement with the literature. Such a reaction, but at later stages, was distinguished for some other plants such as Arabidopsis, maize, *Lupinus albus*, *Stylosanthes*, and others [61,62,63,64]. 

Further GS-MS analysis revealed that the root metabolic profile of young *M. lupulina* plants is a changeable parameter. The profile had just over 300 compounds. The largest group is represented by sugars. The diversity of this group of root metabolites was slightly smaller than in the leaves, whose profile we distinguished earlier [17]. A similar correlation was previously shown for pea plants [25].

The number of investigations devoted to the metabolic profiles of plant roots at early development stages, including that at P deprivation, is very limited. One of the well-documented reactions is the intensification of organic acid synthesis. Accumulation of organic acids is supported with the increase of expression of genes encoding enzymes of TCA [65]. The elevation of the synthesis and further secretion of organic acids from roots lead to an activation of insoluble soil phosphate. According to our data, black medic roots increase the concentration of malic acid at the second week, which was a switch to accumulation of lactic acid with plant aging. We detected a slight increase of glyceric acid. This is known as an important precursor for several phosphorylated compounds such as 2-phosphoglyceric acid, 3-phosphoglyceric acid, and 1,3-bisphosphoglyceric acid. P limitation might result in the accumulation of this acid, as well as AMP. At 14 DAS, roots synthesize threonine, glycine, and proline. All three are known as metabolites that have a significant role in defense against such abiotic stressors as osmoprotectants. Another distinguished reaction under P limitation is the accumulation of sugars. It had earlier been shown that P deficiency can lead to a serious violation of carbohydrate metabolism [42,66]. In black medic roots, we detected the elevation of both fructose and glucose in the roots of the youngest plants. At 24 DAS, a decrement in monosaccharides in roots of non-mycorrhizal black medic plants might be the result of intensive exudation of primary metabolites, including sugars, into the rhizosphere. It possibly includes different mechanisms, such as passive losses and active exudation. Moreover, besides its nutritive role, glucose has important signaling functions [67]. Previously, it has been shown that sugars in young plants perform a regulatory function, intensifying root elongation [68]. Taken together, performed analysis clearly indicates metabolic adaptation of *M. lapulina* roots to P starvation. This adjustment includes different groups of metabolites: carbonic and amino acids, as well as sugars. This reprogramming of plant metabolism coincides with a limitation in the pool of phosphorylated compounds. The most important is that all these effects were detected at the second week of black medic development and varied at a later period with prolongation of P deprivation.

Another well-known mechanism for plants to tolerate P deficiency is to develop arbuscular mycorrhiza: effective symbioses between land plants and fungi. The model object of this study is the highly responsive *M. lupulina* MlS-1 line in which P starvation triggers dwarfism. The AM symbiotic interaction prevents this phenotype from appearing [28,69]. The metabolic background of such reprogramming is not truly understood. It is well documented that AM fungi might consume up to 20% of host plant photosynthates but simultaneously enhance plant growth [1]. Thus, AM implies an intensive exchange of metabolites between symbionts. Of particular interest is discovering such a swap at the earliest stages of host plant development, when metabolites and especially carbohydrates are so necessary for the development of the plant itself. Thereby, we provided a GS-MS analysis of the changes in *M. lupulina* root metabolome after inoculation with the *Rhizophagus irregularis* at 14 and 24 DAS. 

We determined that, already at the earliest investigated period in the roots of young plants, active formation of all symbiotic structures is observed: the formation of appressories on the surface of the roots, as well as arbuscules, vesicles, and intraradical mycelium (Appendix A). At the next stage (24 DAS), the development of these structures was only intensified. In conditions of low P content in the substrate, the functioning of AM led to an acceleration of the development of the aboveground parts of the plant: an increase in the fresh and dry mass of shoots, as well as the number of leaves. This latter is well combined with the previously revealed increase in the level of cytokinins in the shoots of mycorrhized plants, which indicates the stimulating effect of AM on shoot development, mainly through the development of leaves, an organ that provides an increase in photosynthate levels [69].

The appearance of new leaves and an increase in photosynthesis will prompt the enrichment of the metabolic profile of black medic roots. Moreover, it should also be taken into account that the metabolome of mycorrhized roots reflects the metabolic rearrangements of both symbiotic partners. This distinguishes the roots of mycorrhized plants from the leaves, where the age and the development stage have a greater effect on the metabolome than mycorrhization [17]. 

It should be noted that the accumulation of sugars differs significantly in the control and mycorrhized plants. A mycorrhiza-dependent shift in the spectrum of sugars most likely depends also on the intensification of its transport from the leaf and because of its further metabolization by both symbionts in the roots. The intensive alteration at two tested dates (14th and 24th DAS) is illustrated in Figure 3 and Figure 4. This supposition coincides with AM-induced accumulation of “complex sugars”, which are metabolites with sugar residues but distinct from monosaccharides. Such an accumulation of different forms of sugars is a quite common case in many host plants over mycorrhization ([16] and citations therein). The importance of glucose was shown for arbuscule and intraradical fungal development [70] and was supported by the intensification of fungal hexose transporter activity. In addition, among others, we detected trehalose, which is known as a typical fungal metabolite. Accumulation of this disaccharide in AM roots indicated vigorous carbohydrate metabolism of the mycosymbiont [71]. The result supported the idea that sugars from plant shoots are a source of sugar synthesis in AM fungus.

One very interesting fact caught our attention: a decrease in the level of intermediates of the Krebs cycle (citrate, succinate, malate, and fumarate) at the early stages of mycorrhization. A similar decrease in the content of organic acids was demonstrated earlier for dicotyledonous plants but in leaves [6,53]. Organic acids of the TCA cycle such as aconitic acid and fumaric acid reduction in mycorrhizal roots of *M. truncatula* were suggested to be implied with mycorrhiza-induced stimulation of the mitochondrial and plastidial metabolisms [27]. Along with this, the reason for such changes may be related to the inhibition of the early stages of the Krebs cycle, as well as increased transamination reactions, leading to rapid depletion of the ketoacid pool.

Unlike organic acids, the level of a number of amino acids increased during mycorrhization. These include leucine, glycine, histidine, tyrosine, tryptophan, and lysine (Figure 3). Accumulation of a number of amino acids was noted earlier [27,72]. The role of amino acids in the development of the host plant root system is indisputable, especially at such an early stage. At the same time, one more interesting aspect can be considered: the active synthesis of proteins by the mycosymbiont. These secreted effector proteins are supposed to regulate processes in the host plant and are hypothesized to be factors that control symbiotic efficiency and/or host range [73,74,75].

Another important group of primary metabolisms is lipophilic compounds. AM fungi is apparently unable to synthesize a sufficient amount of 16:0 FAs (fatty acids) but has enzymes for further elongation of 16:0 FA to a higher chain length and for FA desaturation [35,76]. Moreover, the application of a visualization approach demonstrated that lipid-producing plastids increase in number and, together with the endoplasmic reticulum, change their position and gather in the neighborhood of arbuscules [36]. *M. truncatula* mutants in AM-specific paralogs of two lipid biosynthesis genes were used to demonstrate the importance of plant lipid biosynthesis for arbuscule development [77]. Unfortunately, our analysis did not reveal an increase in the level of 16:0 and 16:1 FAs in the roots of 14 DAS mycorrhizal plants, which are marker lipids of arbuscular mycosymbionts. However, there was a significant increase in the content of seven different sterols. However, our study did not detect an accumulation of ergosterol, which is suggested as a key factor in arbuscular mycorrhizal fungi growth [37]. Our data coincide with some of the literature which indicates that representatives of the phylum Glomeromycota that form AM contain sterols other than ergosterol [78].

Further development of *M. lupulina* plants led to a significant change in the spectrum of metabolites that reflected the increased role of mycorrhization. The accumulation of amino acids in *M. lupulina* roots increases at 24 DAS, which indicates an upregulation in nitrogen metabolism. Note that accumulation of monosaccharides and the general alteration in the spectrum of carbohydrates was caused by the mycorrhization.

It is no wonder that formation of AM stimulated the accumulation of phosphates in roots both at 14 and 24 DAS. Such an effect was described earlier for leaves of *M. lupulina* [17] and other species, for example, *Sorghum caudatum*, *S. bicolor* [38], *Kennedia coccinea*, *K. stirlingii*, *K. carinata*, *K. prostrata* [79], and others. 

The analysis of the data obtained at different stages of seedling development using the SUS plot (Figure 6) showed the same signs of loadings for most of the metabolites, which confirms the role of mycorrhization in determining root metabolism.

At the final step, we compared the changes, initiated by mycorrhization, in roots and leaves of the *M. lupulina* MlS-1 line at the early stages of its development (a detailed analysis of the metabolic profiles of the leaves was carried out earlier by [17]). Organ specificity was the strongest factor compared to the development stage and even mycorrhization (Figure 7A). We also note a greater variance in the metabolic profiles of leaves in comparison to roots. The reasons of such a difference could be both internal and external. The transformation of energy during photosynthesis determines a wide variety of synthetic processes, which defines a wider range of metabolites and more complicated regulatory responses. In addition, less diversity in the root may be the result of a more constant environment in the soil compared to the air environment of the leaf. Similar conclusions were drawn during an analysis of the metabolomics profiles of *P. sativum* roots and leaves [5,25]. It is shown that the most striking distinguishing feature was the accumulation of amino acids and a number of nitrogen-containing compounds in *M. lupulina* roots (Figure 7B). This was less pronounced in the leaves but intensified with mycorrhization development. While the effect of mycorrhization on the profiles of black medic MIS-1 leaves and roots was quite similar (Figure 7C), AM effects are not always integral and so may represent a high degree of responsiveness on the part of the host plant.

## 4. Materials and Methods 

### 4.1. Plant and Fungus Biomaterials

*Medicago lupulina* line MlS-1, characterized by high mycorrhizal growth response (MGR, AM efficiency), was used to study root metabolome alterations under conditions of low available P in the soil [17]. *Rhizophagus irregularis* effective strain RCAM00320 (Laboratory #4 of Ecology of Symbiotic and Associative Rhizobacteria at All-Russian Research Institute for Agricultural Microbiology, ARRIAM; the strain was previously known as *Glomus intraradices* Shenck&Smith strain CIAM8) was selected as forming highly effective AM symbioses with most agricultural crops [80,81,82] and identified by members of the author’s team [83]. *R. irregularis* is an obligate symbiont of plants, so the culture of AM fungus was grown in *Plectranthus australis* in Laboratory #4 at ARRIAM. Fungal inoculant preparation was described earlier [17]. For inoculation of one *M. lupulina* seedling, a fragment of *P. australis* root with ~100 AM fungal vesicles was used.

### 4.2. Experimental Design and Plant Growth Conditions

Agrochemical soil characteristics: sod-podzolic loam-poor soil with very low P content (P_2_O_5_, 23 mg/kg); K_2_O, 78 mg/kg; organic matter content, 3.64%; pH_KCl_, 6.4; pH_H2O_, 7.3. Soil/sand (2:1) mixture for cultivation was autoclaved at 134 °C, 2 atm. for 1 h with repeated autoclaving after 2 days. Detailed procedures for the experimental design were described earlier [17]. *M. lupulina* seeds were scarified for 5 min in concentrated H_2_SO_4_. Then, the seeds were stratified in Petri dishes for 1 day at +5 °C and then germinated for 2 days at +27 °C in the dark. Seedlings with the same size were grown in a soil–sand substrate. Half of the plants were inoculated with AM inoculant and planted concurrently; the other half was not treated by AM inoculant, as a control. The plants were grown with 4 seedlings in one pot filled with 210 g of soil–sand substrate. Plant watering was carried out every other day up to 0.6 of saturated water content. The protocol for growing using UV-sterilized light phytobox was described earlier [17,69]. The micro-vegetative method provided optimal conditions for AM development and prevented spontaneous infection with rhizobia and other symbiotic microorganisms. Biochemical and microscopic analyses of plants were performed in two stages: (1) on day 14 after sowing and inoculation (DAS), at the stage of development of 1st true leaf in plants with AM and plants without AM; (2) on 24 DAS at the stage of stooling initiation, 3rd leaf development in plants with AM and 3rd leaf initiation in plants without AM. The fresh and dry weight of the roots and aboveground parts of the plants were determined. For subsequent biochemical analysis, the roots of 8 plants were collected for 1 biological repeat (and 3 biological repeats per 1 variant of treatment). These were weighed and quickly frozen in liquid nitrogen and then stored at −80 °C.

### 4.3. Evaluation of Mycorrhization Parameters 

To analyze AM, roots were dried at room temperature and then macerated and stained with trypan blue [84]. Mycorrhization indices were calculated [85]: *M* and *m* were the intensity of root cortex colonization and intensity of colonization in mycorrhized parts of roots, respectively; *a* was the abundance of arbuscules in mycorrhized parts of roots; *b* was the abundance of vesicles in the mycorrhized parts of roots. Microscopic analysis of AM development was carried out using the computer program for calculating mycorrhization indices of plant roots, developed by A.P. Yurkov et al. [86].

### 4.4. Evaluation of Mycorrhizal Growth Response: AM Symbiotic Efficiency

The mycorrhizal growth response (MGR, AM efficiency) was calculated as the fresh (or dry) weight of the aerial parts (or roots), using Odum’s formula:E = ([AM+] − [AM−]) × 100%/[AM−], (1)
where E is the AM symbiotic efficiency (MGR); [AM+] is the value of mycorrhized plant weight; [AM−] is the value of the weight of plants without AM.

### 4.5. GC-MS Analysis 

The sampled mass of 100 mg of roots was collected at 14 and 24 DAS and immediately frozen with liquid nitrogen. Plant materials were ground with a mill (MM 400, Retsch, Germany). Then, metabolites were extracted using 2 ml of extraction mix (methanol/chloroform/water (5:2:1)) and shaken at 900 rpm at 4 °C in a thermoshaker (BioSan TS-100C). Extracts were purified from tissue debris by centrifugation at 12,000 g for 10 min at 4 °C. The supernatant was collected and evaporated in a vacuum evaporator (CentriVap, Labconco, USA). Dried samples were then dissolved and derivatized in pyridine and BSTFA:TMCS 99:1 (Sigma-Aldrich, Burlington, USA) at 90 °C for 20 min. Tricosane was added (normal hydrocarbon) as an internal standard. 

Samples were analyzed with an Agilent 5860 chromatograph equipped with a DB-5 MS capillary column and coupled with an Agilent 5975 quadrupole mass selective (Agilent Technologies, Santa Clara, CA, USA). The flow rate of helium was 1 mL/min. The inlet was operated in splitless mode and temperature was 250 °C. Column temperature regime: initial −70 °C, final −320 °C, rate −5 °C per min. Electron impact ionization was performed at 70 V with an ion source temperature of 230 °C. Analysis of the GC-MS data was processed using the PARADISe software (Department of Food Science Faculty of Science, University of Copenhagen, Denmark, [87]) coupled with NIST MS Search (National Institute of Standards and Technology (NIST), USA). In addition, for deconvolution and metabolite identification, the AMDIS (Automated Mass Spectral Deconvolution and Identification System, NIST, USA) was used. Analytes were identified by mass-spectra and Kovats retention indices by using libraries: NIST2010, Golm Metabolome Database (GMD; [88]). Moreover, an “in house” library was used, which was compiled by the laboratory of analytical phytochemistry with funding BIN RAS # AAAA-A18-118032390136-5.

### 4.6. Statistical Analysis

Statistical analysis of metabolomics data was processed using R 4.2.1 "Funny-Looking Kid" [89]. Data were normalized against the sample median. Outliers were detected and excluded on the basis of Dixon’s test in the *outliers* package [90]. The data were log-transformed and standardized. If a compound was not detected in a sample but was present in the other replicates (minimum 2/3 of replicates), it was considered as a technical error and imputed by KNN (k-nearest neighbors) with *impute* R package [91]. PCA (principal component analysis, PCA) was performed with *pcaMethods* [92]. Orthogonal Partial Least Squares (OPLS-DA) was used for classification with *ropls*. Factor loadings of predictive component and Variable Importance in Projection (VIP) were used to assess the statistical relationship between features and factors of interest [93]. For metabolite set enrichment analysis, the *fgsea* algorithm was used [94].

Metabolite sets for metabolic pathways were downloaded from the KEGG database [95] through *KEGGREST* package [96] with *M. truncatula* as a reference organism. Lists of metabolites for pathways were corrected manually: poorly represented or extra-large sets were excluded, for some metabolites obligatory needful pathways were added, and compounds identified up to class (hexose, disaccharide, among others) were joined to relevant pathways.

To represent metabolites in their biochemical context, a simplified biochemical network was built. For this purpose, metabolites involved in metabolic pathways of reference organisms were revealed from KEGG by *KEGGREST*. Then, the reactions to these compounds were downloaded, and main reaction pairs were extracted as RCLASS. Based on the main reaction pairs, a network was built, which included the 3 shortest paths which were not longer than 5 nodes (paths were extracted with *igraph* package [97]) between all pairs of identified metabolites in profiles. Graphs were built from the *Cytoscape* software [98].

## 5. Conclusions

To sum up the data, we note that, at early stages of AM establishment, the metabolite profiles of *M. lupulina* roots significantly differ in their spectrum from those in the leaves. The roots contain fewer sugars and organic acids. Hence, compounds supporting the growth of mycorrhizal fungus (especially amino acids, the number of lipids, and specific carbohydrates) accumulated and coincided with intensive development in AM structures. This result clearly confirms the intensive development of AM fungus in the roots of young host plants. Such an observation is crucial for determining the further development of *M. lupulina* plants under phosphate deprivation and, in turn, reveals one of the mechanisms of plant adaptation based on symbiosis.

## Figures and Tables

**Figure 1 plants-11-02338-f001:**
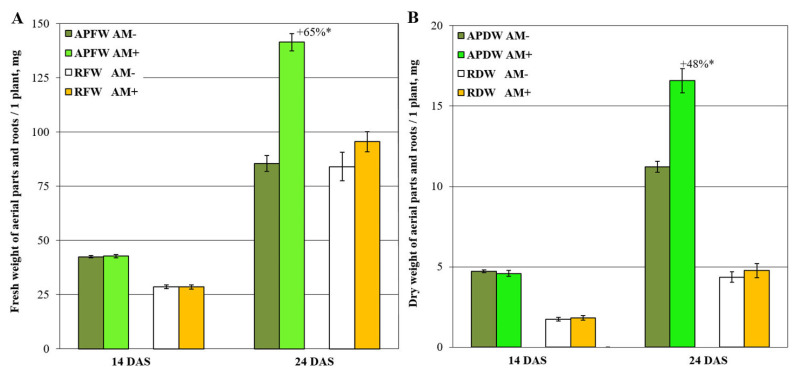
Fresh (**A**) and dry (**B**) weight of aerial parts and roots per one *M. lupulina* plant. “APFW” and “RFW” refer to fresh weight of aerial parts and roots, respectively. “APDW” and “DFW” are the dry weights of aerial parts and roots. “AM−” and “AM+” are the variants without and with AM fungus inoculation. “+65%*” and “+48%*” are significant (*p* < 0.05) mycorrhizal growth response (MGR, AM efficiency) within the same parameter of productivity (ANOVA and Tukey’s test; *p* < 0.05). “DAS” is the day after sowing and inoculation.

**Figure 2 plants-11-02338-f002:**
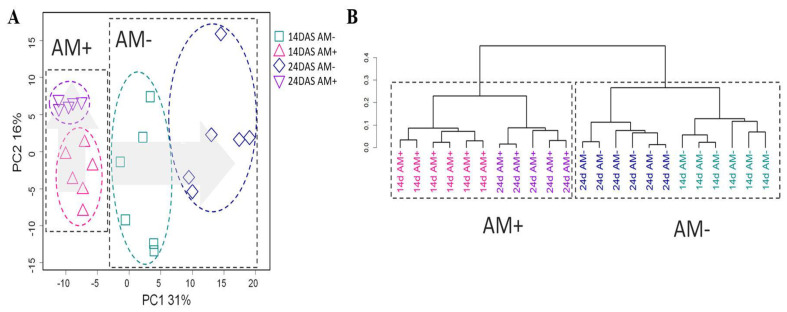
Unsupervised analysis of metabolite profiles of mycorrhized (AM+) and non-mycorrhized (AM−) roots of *Medicago lupulina* plants sampled at 14 and 24 days after sowing. PCA score plot (**A**), dendrogram of hierarchical clustering (**B**).

**Figure 3 plants-11-02338-f003:**
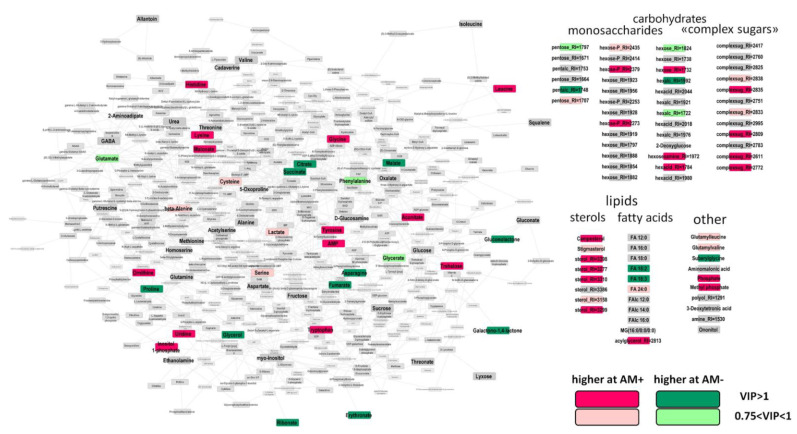
Visualization of differentially accumulated metabolites (DAMs) in response to mycorrhization at 14 DAS as revealed through OPLS-DA. Nodes—metabolites; edges—reactions extracted from the KEGG database (see Section “Materials and Methods”).

**Figure 4 plants-11-02338-f004:**
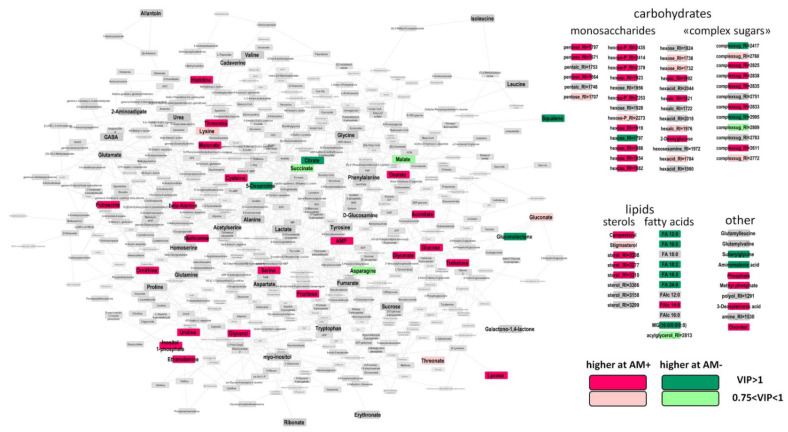
Visualization of differentially accumulated metabolites (DAMs) in response to mycorrhization at 24 DAS revealed through OPLS-DA. Nodes—metabolites; edges—reactions extracted from KEGG database.

**Figure 5 plants-11-02338-f005:**
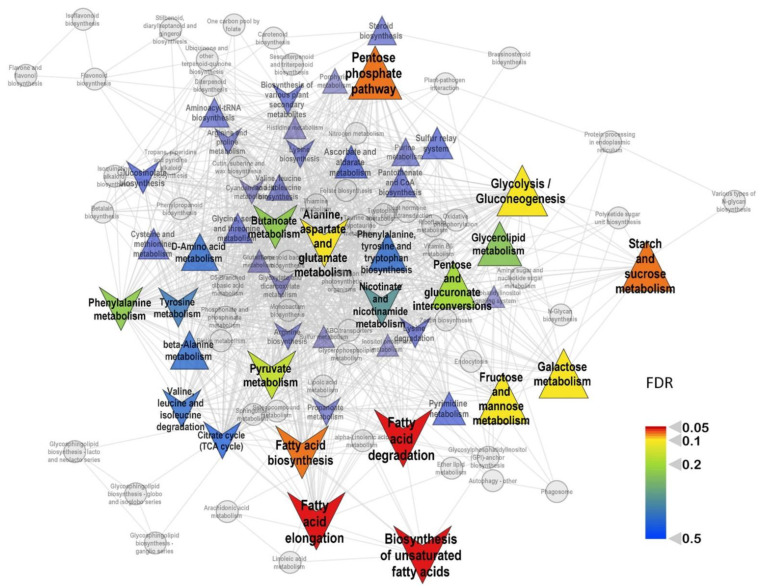
Metabolite sets enrichment analysis based on loadings from OPLS-DA classification of mycorrhized (AM+) and non-mycorrhized (AM−) roots at 24 DAS. Nodes are the paths extracted from KEGG. If paths share metabolites, they are connected by an edge. Color—significance of influence on this pathway; size—strength of influence (|NES|); upward triangles—upregulation at AM+; downward direction—downregulation.

**Figure 6 plants-11-02338-f006:**
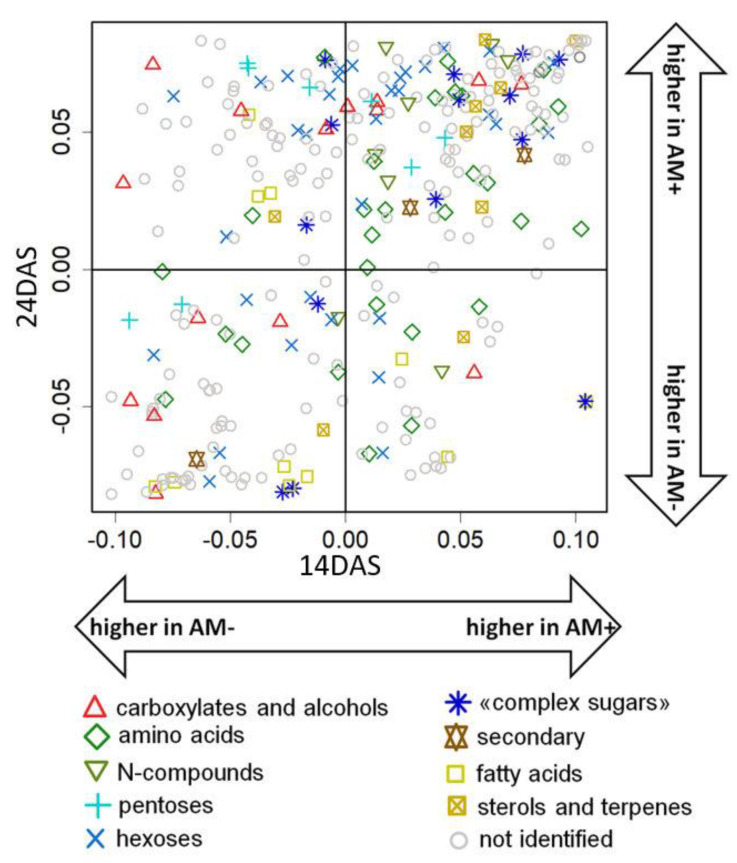
Comparison of mycorrhization effects at 14 and 24 DAS. SUS plot in the space of the loadings from two OPLS-DA models for discrimination of mycorrhized (AM+) and non-mycorrhized (AM−) roots at 14 DAS (abscissa) and 24 DAS (ordinate); positive values correspond to higher content at AM+.

**Figure 7 plants-11-02338-f007:**
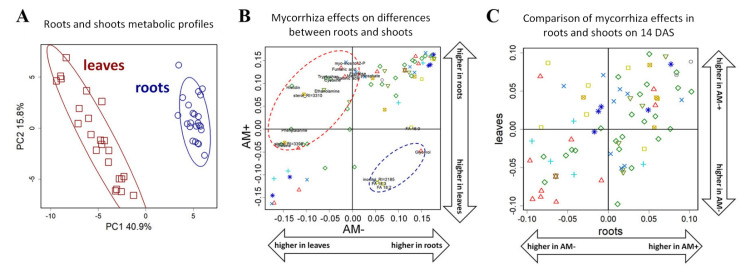
Organ specificity and mycorrhiza effects. Score plot from PCA for metabolites identified in leaves and roots profiles (**A**). SUS plot in the space of loadings from two OPLS-DA models for discrimination of roots and leaves in AM+ and AM− plants (**B**). SUS plot in the space of the loadings from two OPLS-DA models for discrimination of AM+ and AM− roots (abscissa) and shoots (ordinate) of plants at 14 DAS (**C**). Symbols for carboxylates and alcohols, amino acids, N-compounds, pentoses, hexoses, «complex sugars», secondary metabolites, fatty acids, sterols and terpenes, not identified compounds are the same as in Figure 6.

## Data Availability

Not applicable.

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
