# Peer review of "The Role of Medicago lupulina Interaction with Rhizophagus irregularis in the Determination of Root Metabolome at Early Stages of AM Symbiosis"

_plants, 2022, doi:10.3390/plants11182338_

Round 1

Reviewer 1 Report

The paper by Yurkov et al examines the changes in the metabolome of a type of clover (Medicago lupulina) in response to plant age (with two sampling dates) and mycorrhization, in conditions of phosphorous deficit. The authors did a nice experimental work, but I think that the manuscript still has shortcomings that need to be appropriately addressed. My main comments are:

1. The use of English should be improved. There are many grammatical and syntactical errors but, more importantly, the meaning of many sentences is obscure and it is very difficult to extract a clear message from the text.

2. The introduction and the discussion are too long, going over many data that are irrelevant and unfocusing the paper. In both, authors go in great detail over other examples of mycorrhization that, often, seem to have little connection to the main subject of the paper. The detailed description of many examples could be appropriate for a review, but in this manuscript make reading difficult.

3. It is not clear what are the novelty and contributions of this paper. From the text, it seems that other authors have done very similar experiments with this and other plant species, and it is not clear what does this paper contribute to the advancement of knowledge.

4. There is a general lack of interpretation of data. The authors did a great deal of work to obtain a huge amount of data on the metabolome of plants; however, for the reader it feels like the authors present more or less a shopping list of metabolites. Although it seems clear that mycorrhization has a stronger effect than plant age on the types of metabolites produced by the symbiosis, there are no further analyses of the data. I think authors could discuss and integrate the variations on the metabolome and their possible impact on plant development, specially considering how the mycorrhization improves plant growth. This would be important to understand how and why the fungus stimulates plant development. Additionally, authors could discuss the relevance and impact of their data; since M. lupulina is a plant highly responsive to symbiosis, it would be appropriate to discuss how could their data apply to other plant species.

Author Response

Dear Reviewer.

We are very thankful for the positive attitude to our investigation and foryour critical comments. We seriously rewrote “Introduction” and “Discussion” of the manuscript, and we believe that the manuscript has now more detailed analysis of cited literatureand obtained data.

Point 1: The use of English should be improved. There are many grammatical and syntactical errors but, more importantly, the meaning of many sentences is obscure and it is very difficult to extract a clear message from the text.

Response 1: Text of the manuscript was corrected by a native English speaker after we have finished all needed text corrections.

Point 2:  The introduction and the discussion are too long, going over many data that are irrelevant and unfocusing the paper. In both, authors go in great detail over other examples of mycorrhization that, often, seem to have little connection to the main subject of the paper. The detailed description of many examples could be appropriate for a review, but in this manuscript make reading difficult.

Response 2: We agree with this comment of honorable reviewer. We have reorganized citation and shortened Introduction to avoid “obscuring” of the main objective of the investigation.

Point 3: It is not clear what are the novelty and contributions of this paper. From the text, it seems that other authors have done very similar experiments with this and other plant species, and it is not clear what does this paper contribute to the advancement of knowledge.

Response 3: The novelty and contribution of this paper can be expressed as following:

  • we focused on elucidation of arbuscular mycorrhiza effect on host plant root metabolome. The peculiarity of the investigation is the model object - the selected highly responsive (up to MGR>350%) Medicago lupulina line MlS-1. Mycorrhization was established with highly effective strain RCAM00320 of Rhizophagus irregularis AM fungus under conditions of low P level in the substrate.
  • we pay special attention on metabolic alterations at the very early stages of development - the 1st and 2nd Root metabolic profile is a sum of host plant itself and actively developing AM fungi. Plant is very sensitive at this stage to any limitation in photosynthates and macro/micro-nutrients. We tried to get the knowledge which of these factors would be the most crucial to determine root metabolome. This question of a special importance because previous similar investigations were provided with plant model objects with MGR below 100% in most cases.
  • Moreover previously published data were obtained at later stages of plant development and the period of determination was usually calculated as days after inoculation (dai). This makes it quite difficult to reveal priority of mechanisms controlling AM establishment and plant growth.

Taken together we can assume that our investigation provided an important data representing new knowledge on establishment of plant-fungi symbiosis at metabolic level.

Point 4: There is a general lack of interpretation of data. The authors did a great deal of work to obtain a huge amount of data on the metabolome of plants; however, for the reader it feels like the authors present more or less a shopping list of metabolites. Although it seems clear that mycorrhization has a stronger effect than plant age on the types of metabolites produced by the symbiosis, there are no further analyses of the data. I think authors could discuss and integrate the variations on the metabolome and their possible impact on plant development, specially considering how the mycorrhization improves plant growth. This would be important to understand how and why the fungus stimulates plant development. Additionally, authors could discuss the relevance and impact of their data; since M. lupulina is a plant highly responsive to symbiosis, it would be appropriate to discuss how their data could apply to other plant species.

Response 4: Data of our investigation is fully represented in Table S1 and it is quite full described in “Results”. This may help to readers to provide their own comparison. According to reviewer suggestion we spent efforts to improve the interpretation of obtained results in “Discussion”. We added a special section on metabolic alterations in plant roots under P deprivation. Unfortunately data on such alterations in young plants and/or seedlings is very limited.

We also rewrote and focused on alterations during mycorrhization. Our model is quite different from previously published because selected line of M. lupulina is very sensitive to symbiosis with AM fungi. We believe that this model highlighted the effect of mycorrhization. Thus it might be considered as revealing the background of AM effectiveness. Such pronounced metabolic alterations makesit important further transcriptomic and proteomic approach. We have already started the research on the analysis of the transcriptome of leaves and roots of analyzed highly effective M. lupulina line MlS-1.

Sincerely yours,

Authors

Reviewer 2 Report

In the manuscript, the authors highlight the importance of plant-fungi active metabolic interaction at the early stages of host plant development for the determination of symbiotic efficiency. Overall, the manuscript is well written I have small concerns related to the points mentioned below:

The introduction is relatively poor.

The discussion is poor as well. 

Author Response

Dear Reviewer,

We appreciate such a positive attitude to our manuscript. We have tried to improve both Introduction and Discussion. We made it more clearly our working hypothesis and focuses on specificity of revealed metabolic alterations. We also restructured Conclusion. We hope that our efforts improve the manuscript and it is ready now for publication.

Sincerely yours,

Authors

Round 2

Reviewer 1 Report

The revised version of the article by Yurkov et al has definitely improved. The introduction and results now read  significantly better, although it seems the proofreader is not a scientist and there are still sentences that do not make sense (e.g. lines 133-134). The introduction could still be shortened with little effort, to condense sentences (e.g. lines 94 to 100 could be shortened to "from 42 to 180 dai", and it would be much clearer). Also, in general, sentences are very convoluted and long, instead of going directly to the point, which obstruct reading. Nevertheless, I still have a few comments that I feel are unresolved:

1. Discussion has been partially rewritten, but has not been revised by an English speaker. It is unjustifiably long, very hard to read and very descriptive, without much discussion of the data. Language is also very convoluted and ideas are at times difficult to understand

2. Is not easy to see what are the novelty and contributions of this paper, and this is not evident from the discussion.

3. The discussion seems to still be very descriptive, as an enumeration of compounds; therefore, the paper lacks an integration of the data to formulate hypothesis to explain the benefit of the symbiosis. As said in the first review, authors could discuss the relevance and impact of their data; since M. lupulina is a plant highly responsive to symbiosis, it would be appropriate to discuss how could their data apply to other plant species.

4. There is no indication of the number of repetitions or the number of plants used for each experiments. These data should be in the Materials and methods section and, desirably, in figure/Table legends. For instance, in Fig 1 you should state that bars are the average weight of whatever number of plants, and that the experiment was repeated whatever number of times.